# Cross-talk between red blood cells and plasma influences blood flow and omics phenotypes in severe COVID-19

Steffen M Recktenwald[1]*[†], Greta Simionato[1,2]*[†], Marcelle GM Lopes[1,3], Fabia Gamboni[4], Monika Dzieciatkowska[4], Patrick Meybohm[5], Kai Zacharowski[6,7], Andreas von Knethen[6,7], Christian Wagner[1,8], Lars Kaestner[1,9], Angelo D'Alessandro[4], Stephan Quint[1,3]

[1]Dynamics of Fluids, Department of Experimental Physics, Saarland University, Saarbrücken, Germany; [2]Institute for Clinical and Experimental Surgery, Campus University Hospital, Saarland University, Homburg, Germany; [3]Cysmic GmbH, Saarbrücken, Germany; [4]Department of Biochemistry and Molecular Genetics, University of Colorado Denver, Aurora, United States; [5]Department of Anesthesiology, Intensive Care, Emergency and Pain Medicine, University Hospital Wuerzburg, Wuerzburg, Germany; [6]Department of Anesthesiology, Intensive Care Medicine and Pain Therapy, University Hospital Frankfurt, Frankfurt, Germany; [7]Fraunhofer Institute for Translational Medicine and Pharmacology ITMP, Frankfurt, Germany; [8]Department of Physics and Materials Science, University of Luxembourg, Luxembourg City, Luxembourg; [9]Theoretical Medicine and Biosciences, Campus University Hospital, Saarland University, Homburg, Germany

*For correspondence:
steffen.recktenwald@uni-saarland.de (SMR);
greta.simionato@uni-saarland.de (GS)

[†]These authors contributed equally to this work

**Abstract** Coronavirus disease 2019 (COVID-19) is caused by the Severe Acute Respiratory Syndrome Coronavirus 2 (SARS-CoV-2) and can affect multiple organs, among which is the circulatory system. Inflammation and mortality risk markers were previously detected in COVID-19 plasma and red blood cells (RBCs) metabolic and proteomic profiles. Additionally, biophysical properties, such as deformability, were found to be changed during the infection. Based on such data, we aim to better characterize RBC functions in COVID-19. We evaluate the flow properties of RBCs in severe COVID-19 patients admitted to the intensive care unit by using microfluidic techniques and automated methods, including artificial neural networks, for an unbiased RBC analysis. We find strong flow and RBC shape impairment in COVID-19 samples and demonstrate that such changes are reversible upon suspension of COVID-19 RBCs in healthy plasma. Vice versa, healthy RBCs resemble COVID-19 RBCs when suspended in COVID-19 plasma. Proteomics and metabolomics analyses allow us to detect the effect of plasma exchanges on both plasma and RBCs and demonstrate a new role of RBCs in maintaining plasma equilibria at the expense of their flow properties. Our findings provide a framework for further investigations of clinical relevance for therapies against COVID-19 and possibly other infectious diseases.

## Editor's evaluation

This report illustrates a comprehensive account detailing the marked alteration of red blood cell (RBC) morphology that occurs with COVID-19 infection. A particularly important result is the observation that RBC morphology is dramatically affected by plasma from COVID-19 patients and reversible with plasma from healthy donors. The claims of the manuscript are well supported by the data, and the approaches used are thoughtful and rigorous. The results are important for consideration

of the broader pathophysiology of COVID-19, particularly with regard to the impact on vascular biology and will be of interest to the readership of *eLife*.

## Introduction

Infection with the Severe Acute Respiratory Syndrome Coronavirus 2 (SARS-CoV-2) can lead to the development of the Coronavirus disease 2019 (COVID-19) (*Wu et al., 2020*). The pathophysiology of COVID-19 is characterized by respiratory manifestations but also involves increased inflammatory responses (*Mehta et al., 2020*), alterations in the number and phenotype of blood cells (*Mann et al., 2020*; *Kubánková et al., 2021*), microthrombotic complications, and vascular occlusions that can be fatal (*Tang et al., 2020*; *Varatharajah and Rajah, 2020*; *Della Rocca et al., 2021*). In these changes, red blood cells (RBCs) were found to exhibit structural protein damage, membrane lipid remodeling (*Thomas et al., 2020a*), dysregulation in serum levels of coagulation factors (*D'Alessandro et al., 2020*), as well as altered physical and rheological properties, such as shape, size, and deformability (*Kubánková et al., 2021*; *Renoux et al., 2021*). Changes in RBC mechanical properties, such as its bending rigidity and cytoplasm viscosity, can dramatically alter their morphology, reduce microvascular perfusion, damage the endothelial cells lining the blood vessel walls, and impact flow behavior and hemorheology of blood in the circulatory system, eventually affecting gas transport efficiency (*Lipowsky, 2005*; *Matthews et al., 2015*; *Lanotte et al., 2016*; *Di Carlo, 2012*; *Piety et al., 2021*; *Caruso et al., 2022*). Such effects may play a role in COVID-19 severity. However, the origin of pathological changes of COVID-19 RBCs and their impact on blood flow remains poorly understood. RBC flow properties are particularly relevant in the microvasculature, especially in capillaries, where gas exchanges and microthrombotic events occur. Recently, *Kubánková et al., 2021* studied the microfluidic flow of RBCs from COVID-19 patients suspended a in phosphate-buffered saline (PBS) solution in a channel with a cross-section of $20 \times 20~\mu m^2$, resulting in cell velocities up to $30~cm~s^{-1}$. At such high velocities, RBC deformability turned out to be heterogeneous, with most of the RBCs being strongly elongated in the flow direction but some exhibiting circular, less deformable shapes compared with healthy controls. In human capillaries, which have diameters similar to RBC size (about 10 μm), confined flows confer to RBCs characteristic shapes that depend on the cell velocity and the surrounding medium (*Guckenberger et al., 2018*; *Recktenwald et al., 2022a*; *Recktenwald et al., 2022b*). We previously used a microfluidic setup with channel dimensions similar to the RBC size and analyzed RBC shapes at high detail by means of artificial neural networks (*Kihm et al., 2018*) in a velocity range that resembles microvascular in vivo flow. Flow properties of RBCs depend on biophysical cell features that are influenced by biochemical characteristics. In order to elucidate the effects of COVID-19 on RBCs, we study their shapes and flow properties in autologous and allogeneic plasma in similar confined microfluidic channels in comparison to healthy controls. We observe RBC flow impairment in COVID-19 plasma and a full recovery upon suspension in healthy plasma. To better understand these effects, we additionally characterize both RBCs and plasma content by proteomics and metabolomics analyses, highlighting a mutual influence between RBCs and plasma and discovering a new role of RBCs in influencing plasma homeostasis in COVID-19.

## Results

### COVID-19 patients exhibit pathological changes of RBC shapes in capillary flow

We investigate the microfluidic flow behavior of RBCs in autologous as well as in allogeneic plasma. Therefore, patient RBCs are suspended in blood group-matching control plasma and control RBCs in patient plasma. Hence, we obtain four sample groups; (i) control RBCs in control plasma (CinC), (ii) patient RBCs in patient plasma (PinP), (iii) control RBCs in patient plasma (CinP), and (iv) patient RBCs in control plasma (PinC) (*Figure 1A*). RBCs of the four sample groups are imaged in stasis for the assessment of shapes and cluster formation at rest. While healthy controls (CinC) form biconcave RBCs that organize into rouleaux, sphero-echinocytes for COVID-19 patients in autologous plasma (PinP) are found in stasis and highlight impaired RBC clustering (*Figure 1B*, *Figure 1—figure supplement 2*). However, upon plasma exchange (PinC), COVID-19 RBC shapes in stasis revert to biconcave disks

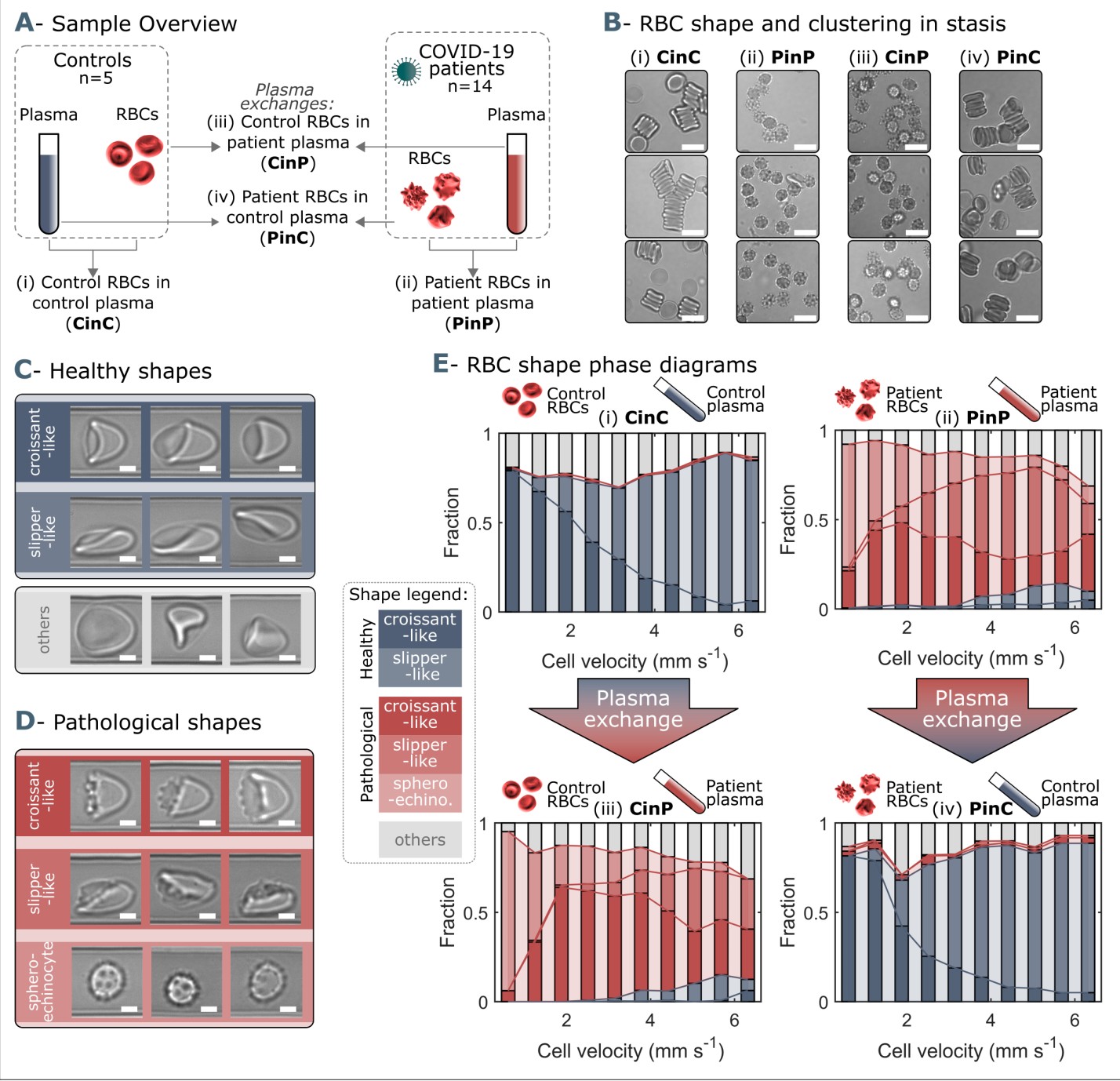

**Figure 1.** Representative red blood cell (RBC) shapes in healthy and COVID-19 patients and related phase diagrams. (**A**) Overview of the four sample groups, referred to as 'X RBCs in X plasma', e.g. CinC for Control RBCs in Control plasma, and so on for PinP, PinC, and CinP. (**B**) Representation images of RBC shapes in stasis for the four sample groups. Scale bars in (**B**) correspond to a length of 10 μm. Representative images of (**C**) healthy and (**D**) pathological RBC shapes found in COVID-19 patients. Scale bars in (**C**) and (**D**) correspond to a length of 2.5 μm. (**E**) Representative shape phase diagrams of two donors (a control and a patient) for the four sample groups. The upper panels show the phase diagrams of the donors in autologous plasma, while the lower panels correspond to the same donors in allogeneic exchanged plasma.

The online version of this article includes the following source data and figure supplement(s) for figure 1:

**Source data 1.** Patient information.

**Figure supplement 1.** Single-cell flow characteristics.

**Figure supplement 2.** Brightfield microscopy and quantification of RBC shapes and clusters in stasis.

and are able to aggregate into rouleaux. The opposite trend is observed for control cells suspended in patient plasma (CinP). The ratio of RBCs in clusters over the total number of RBCs quantifies the differences between CinC and PinP and shows the recovery of patient RBCs for PinC (*Figure 1—figure supplement 2*, left panel). The number of RBCs per cluster in CinC is about twice the number of PinP. Notably, patient cell shape and rouleaux formation recovery in control plasma (PinC) results in no significant differences with control RBCs (CinC) (*Figure 1—figure supplement 2*, right panel).

The RBC suspensions are pumped through microfluidic channels mimicking capillary flow conditions and the RBC velocity, lateral cell position in the y-direction, and projection area of each cell are analyzed (*Figure 1—figure supplement 1*). In confined flows with dimensions on the same order of magnitude as the diameter of RBCs, healthy RBCs predominantly deform into centered croissants and off-centered slipper shapes (*Figure 1C*), depending on the cell velocity. RBCs of COVID-19 patients additionally exhibit complementary pathological croissant and slipper shapes that resemble their corresponding healthy classes but display pronounced spicules on the membrane (*Figure 1D*). Further, COVID-19 patients show sphero-echinocytes in flow, known as irreversible transformations of RBCs that can occur upon ATP depletion, high pH, or exposure to anionic detergents (*Lim, 2009*; *Bernhardt and Ellory, 2013*; *Figure 1D*). The so-called RBC shape phase diagram (*Kihm et al., 2018*; *Guckenberger et al., 2018*; *Recktenwald et al., 2022a*; *Recktenwald et al., 2022b*) describes the frequency of occurrence for RBC morphologies as a function of cell velocity. While croissant- and

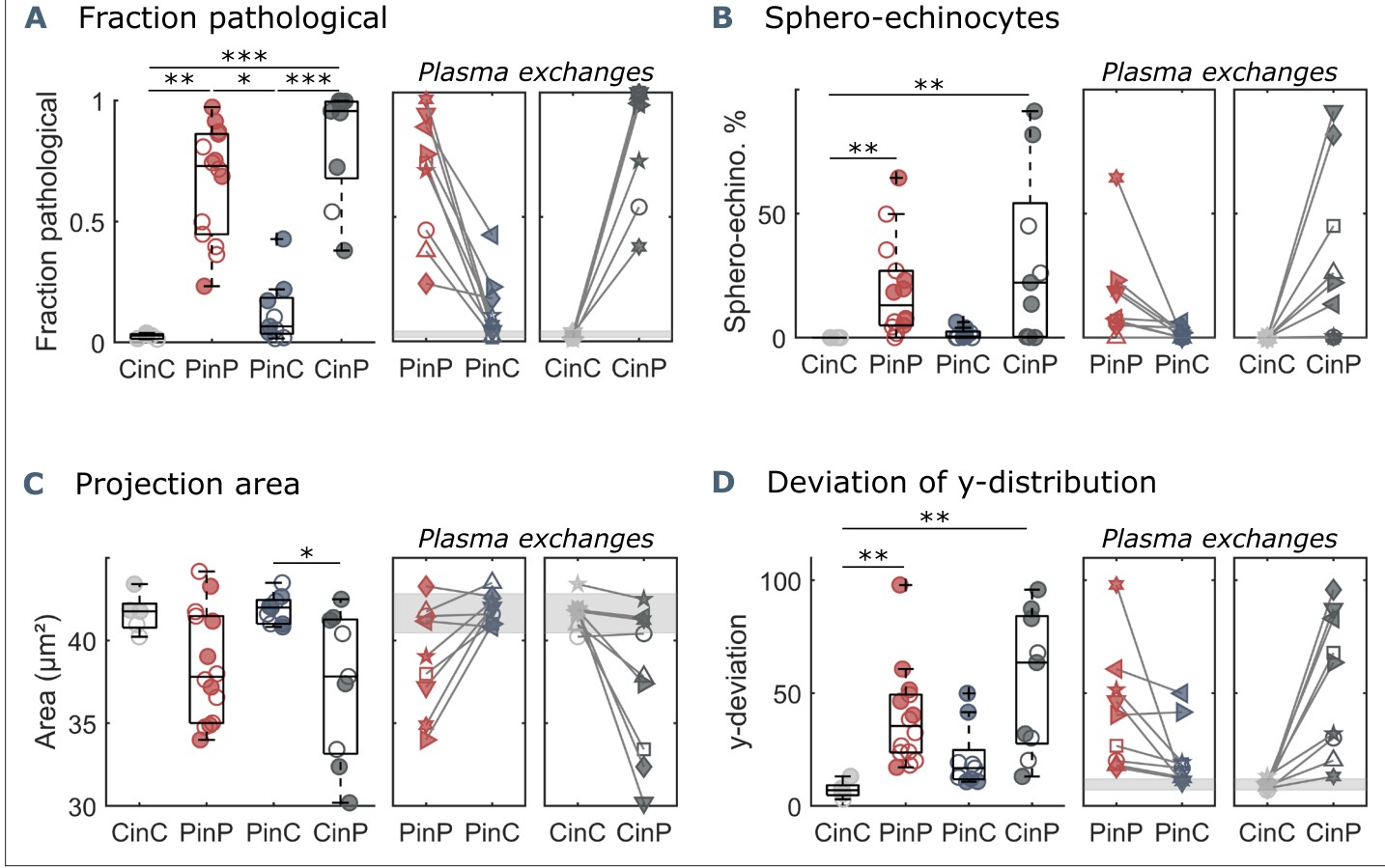

**Figure 2.** Parameters used for microfluidic flow analysis for the four sample groups. (**A**) Fraction of pathological RBC shapes, (**B**) percentage of sphero-echinocytes, (**C**) RBC 2D projection area in a velocity range of 1-3 mm s$^{-1}$, and (**D**) deviation of the RBC distribution in y-direction based on the average of controls (CinC) in a velocity range of 5-10 mm s$^{-1}$. Left panels (**A–D**) show data as boxplots with superimposed individual data points. Filled symbols correspond to samples that are included in the metabolomics and proteomics analyses (*Figure 3* and *Figure 4*). The bottom and top of each box are the 25th and 75th percentiles of the sample, respectively. The line in the middle of each box is the sample median. Whiskers go from the end of the interquartile range to the furthest observation. Data beyond the whisker length are marked as outliers with '+' signs. * refers to a significance level of $p<0.05$, ** to $p<0.01$, and *** to $p<0.001$. Right panels (**A–D**) show the effect of plasma exchange for individual donors. Gray areas correspond to the mean and standard deviations for controls (CinC) of the corresponding data.

slipper-like shapes emerge for control samples in autologous plasma (CinC), corresponding pathological shapes and sphero-echinocytes dominate for patient RBCs in autologous plasma (PinP) (*Figure 1E*, top graphs). This initial assessment highlights the dramatic shape difference between control and COVID-19 patient RBCs. However, upon plasma exchange with healthy control plasma, patient RBCs (PinC) show a shape reversal, exhibiting a phase diagram similar to healthy controls. In contrast, control RBCs in allogeneic exchanged COVID-19 plasma (CinP) result in a drastic RBC shape deterioration (*Figure 1E*, bottom graphs).

In microcapillary flow, the fraction of pathological cells, defined as the number of pathologically shaped cells (*Figure 1D*) divided by the total number of pathological and healthy cells (*Figure 1C*), quantifies the extent of pathological RBC shape changes. While this fraction is considerably small for healthy controls (CinC), COVID-19 patients (PinP) show pathological fractions of up to one in autologous plasma (*Figure 2A*). Upon plasma exchange (PinC), all patients exhibit a decrease in the pathological fraction, hence a reduction in the number of pathological RBC shapes. In contrast, healthy RBCs suspended in COVID-19 plasma (CinP) show an increase in pathological shapes compared to their control condition (CinC). Similarly, suspension in allogeneic plasma results in a reduction of sphero-echinocytes for all patients, while an increase is observed for the controls in COVID-19 plasma (*Figure 2B*). The pronounced number of sphero-echinocytes in RBCs suspended in COVID-19 plasma leads to a decrease of the 2D projection area in the x-y-plane compared to healthy shapes (*Figure 2C*). The large number of sphero-echinocytes and the impaired deformability of pathological RBCs in COVID-19 plasma (PinP and CinP) hinders the formation of stable slipper-shaped RBCs at high velocities. This causes pronounced deviations from the single-cell flow behavior of healthy cells, as expressed by the deviation in the equilibrium position in y-direction (*Figure 2D*).

## Multiomics analyses reveal a tight interaction between plasma and RBCs

To delve into the molecular underpinnings of RBC shape changes in COVID-19 and the reversal of such phenomena upon incubation with healthy control plasma, we perform multiomics analyses, including proteomics and metabolomics, of the four sample groups (CinC, PinP, PinC, and CinP). RBCs and plasma from all four conditions are investigated separately and the top 50 metabolites and proteins, which were previously found to be altered in COVID-19 samples and that exhibited the most pronounced changes, are examined.

### RBCs affect plasma content

The comparison between control (CinC) and patient (PinP) plasma shows two clusters of differentially abundant proteins and metabolites (*Figure 3A*). Control plasma content results in higher levels of albumin (ALB), transferrin (TF), and gelsolin (GSN). Decreased levels of such proteins are associated with hypercoagulability, higher inflammatory state, and severity in COVID-19, respectively (*Violi et al., 2021*; *Claise et al., 2022*; *Messner et al., 2020*). On the contrary, patient plasma is associated with an increase in inflammation and coagulation markers, such as complement cascade proteins (such as C2, C3, C5, C7, C9), SERPINA1, SERPINA3, and SERPINA G1 and C-reactive protein (CRP).

After placing RBCs of patients in control plasma (PinC) and vice versa control RBCs in patient plasma (CinP), plasma contents immediately change, influenced by RBCs. Specifically, patient plasma becomes comparable to control plasma content upon suspension with control RBCs (CinP), while the opposite effect occurs in control plasma suspended with patient RBCs (PinC) (*Figure 3A*). This shows a strong influence of RBCs on plasma content and highlights the establishment of an equilibrium between RBCs and plasma. Notably, the fraction of pathological RBC shapes and sphero-echinocyte percentage are comparable in RBCs suspended in patient plasma, although plasma content is different. This means that RBC influence on plasma content occurs at the expense of RBC morphology, which in turn, affects RBC flow properties. Additionally, these results show that the impairment of RBC shape is not directly caused by a specific plasma component but may derive from a complex interaction between plasma and RBCs. In agreement with the data in the heat map, principal component analysis (PCA) highlights four distinct groups, showing that plasma contents of the groups CinP and CinC result in neighboring clusters, while PinC and PinP clusters strongly overlap (*Figure 3B*).

Within the identified significantly different molecules in plasma (*Figure 3A* heatmap and *Figure 3—source data 1*) tryptophan metabolites deserve consideration. As previously described (*Thomas

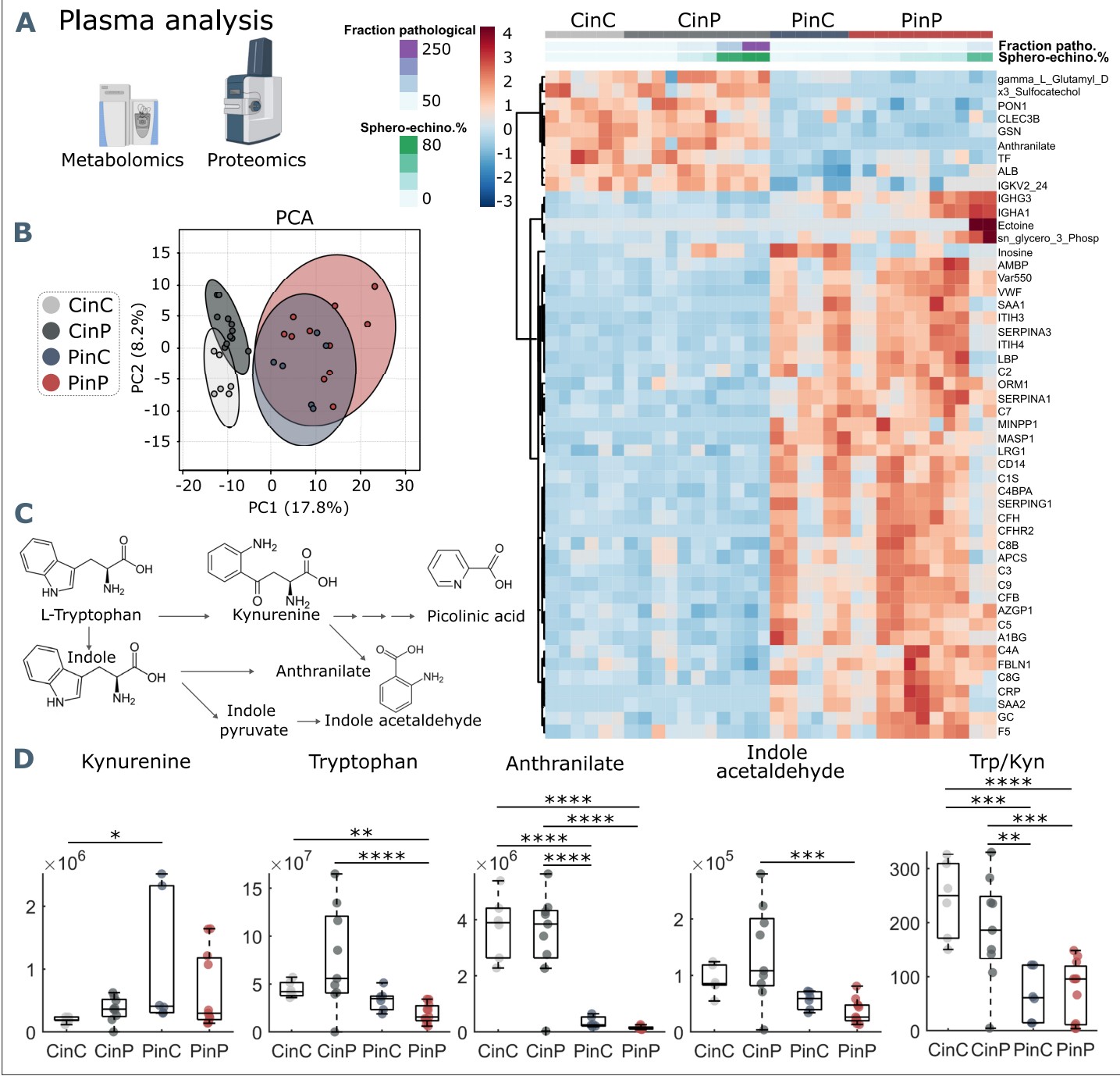

**Figure 3.** Metabolomics and proteomics analyses of plasma from the four sample groups. (**A**) Hierarchical clustering analysis of the top 50 metabolites and proteins by two-way ANOVA test that exhibited the most pronounced changes. (**B**) Principal component analysis (PCA) performed on the metabolomics and proteomics plasma data presented in *Figure 3—source data 1* shows distinct groups. (**C**) Schematic representation of tryptophan pathway to kynurenine, anthranilate, and indole acetaldehyde formation. (**D**) Boxplots of all samples for each group showing statistical differences in selected molecules generated in the tryptophan-kynurenine pathway. Y-axis indicates peak areas for each selected analyte, as determined by UHPLC-MS. Statistically-significant differences exist between controls (CinC) and patients (PinP) in the levels of tryptophan and anthranilate. Plasma exchanges cause strong significant differences in anthranilate levels and the ratio tryptophan-kynurenine (Trp/Kyn) in all groups, a marker of COVID-19 disease severity and mortality in previous studies (***D'Alessandro et al., 2021b***).

The online version of this article includes the following source data for figure 3:

**Source data 1.** All analyzed metabolites and proteins.

*et al., 2020b*), tryptophan metabolism is altered in COVID-19 plasma, resulting in decreased anthranilate and indole acetaldehyde and increased kynurenine in patients, which is associated with a higher mortality risk (*Thomas et al., 2020b*; *Mangge et al., 2021*; *Figure 3C*). Compared to control conditions (CinC), plasma kynurenine significantly increases upon exposure of patient RBCs in control plasma (PinC). Tryptophan levels are significantly lower in patient plasma (PinP) compared to controls (CinC), while they significantly increase for control RBCs in patient plasma (CinP). Plasma anthranilate levels and tryptophan/kynurenine (Trp/Kyn) ratio significantly differ in the groups with control RBCs (CinC and CinP) compared to patients groups (PinC and PinP). These data suggest a role of RBCs in buffering kynurenine content and related metabolites, thus potentially protecting from an increased risk for mortality in COVID-19 patients (*Figure 3D*).

## Plasma components affect RBC content

Omics analyses on RBCs from each group reveal a cluster of proteins and metabolites that is comparable between CinC and PinP. Most of them involve plasma proteins, such as albumin (ALB), fibrinogen (FGA and FGB), gelsolin (GSN), transferrin (TF), serpins (SERPINA1, SEPRINC1), and immunoglobulin variables (IGHV439, IGHG2, IGKC) (*Figure 4A*), which result significantly different in the respective control and patient plasma samples.

Additionally, two clusters indicate that some characteristics of RBCs are maintained only when they are suspended in autologous plasma. Control RBCs (CinC) show higher levels of pantothenol, which is known to have antiseptic properties (*Saliba et al., 2005*), adenosine and hydroxyglutarate that were found decreased in hypoxia (*Nemkov et al., 2018*), and formyl-kynurenine, probably buffering plasma kynurenine (*Figure 4A*). In contrast, the most abundant proteins in patient RBCs (PinP) are linked to cellular responses to stress (*Figure 4—figure supplement 1*), involving ubiquitination and protein degradation by the proteasome, which may be related to a higher oxidative stress. Increasing levels of guanidinoacetate, ornithine, urate, and L-citrulline indicate higher catabolism of arginine.

Since RBCs influence plasma metabolites and protein abundance, they must necessarily adapt their metabolism and protein content. Indeed, an effect occurs when RBCs are suspended in allogeneic plasma (CinP and PinC), resulting in changed levels of plasma proteins (ALB, FGA, FGB, GSN, TF, SERPINA1, SERPINC1) and immunoglobulin variables (IGHV439, IGHG2, IGKC) (*Figure 4A*). These data indicate RBC acute ability to equilibrate their protein levels when coming in contact with new plasma, since the first cluster is comparable between CinC and PinP and between CinP and PinC (*Figure 4A and B*). However, the changes in RBC metabolism and protein content that occur upon suspension with allogeneic plasma do not reflect RBC morphological changes, since PinP samples exhibit many more pathological RBC shapes than PinC samples.

## Pathological RBC shapes are associated with markers of inflammation, oxidation, and hypoxia

### RBC morphological correlations with plasma components

The main plasma components correlating with sphero-echinocytes are lactate and 2,3-diphospho-glycerate (2,3-DPG), associated to increased glycolysis; nicotinamide and tryptophanamide, related to tryptophan metabolism; glutamate, creatine, and hypoxanthine, which was previously found to be positively correlated with creatinine (*Thomas et al., 2020b*); lactoferrin (LTF), an iron-binding protein with antimicrobial and anti-inflammatory activity (*Conneely, 2001*); ectoine a compound associated to anti-inflammatory properties (*Bownik and Stępniewska, 2016*) and found in COVID-19 sera, as well as 2-oxoglutarate (also known as $\alpha$-ketoglutarate) (*Kaur et al., 2021*; *Figure 4C*). The fraction of pathological cells also correlates to hypoxanthine, ectoine, and nicotinamide. TF levels negatively correlate to sphero-echinocyte percentage and fraction of pathological RBC shapes. TF was previously reported to decrease during COVID-19 infection (*Claise et al., 2022*). Similarly, the levels of arginine and tyrosine were previously reported to negatively correlate with disease severity, as gleaned by the levels of the inflammatory cytokine interleukin-6 (IL-6) (*Thomas et al., 2020b*). Analogously, decreases are observed in the levels of sulfur-containing amino acids, thiocysteine, 3-sufinoalanine, and cystine, a marker of redox homeostasis.

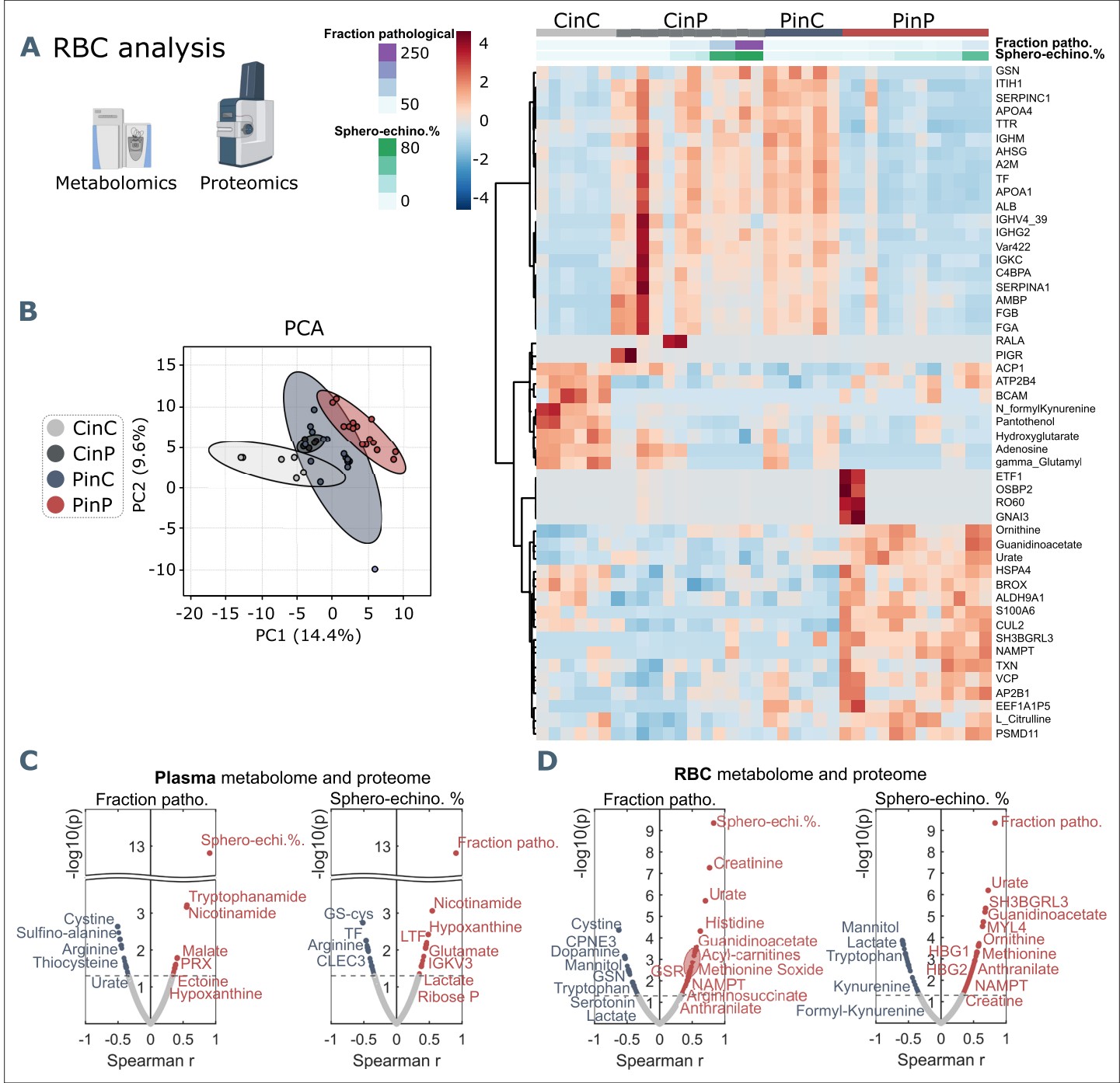

**Figure 4.** Metabolomics and proteomics analyses on RBCs of the four sample groups and correlation analyses with shape parameters. (**A**) Hierarchical clustering analysis of the top 50 metabolites and proteins in RBCs by two-way ANOVA test that exhibited the most pronounced changes. (**B**) PCA analysis performed on the metabolomics and proteomics from RBC content presented in *Figure 3—source data 1* showing clusters of each sample group, where CinC is neighboured by the overlapping clusters CinP and PinC, and PinP as the furthest cluster. Plasma and RBC Spearman correlation analysis of RBC shape parameters with omics data. Volcano plot representations highlight the most significant proteins and metabolites in plasma (**C**) and in RBCs (**D**) positively (red) or negatively (blue) correlated to the fraction of pathological RBC shapes (left panels in **C** and **D**) or sphero-echinocyte percentage (right panels in **C** and **D**). p values are plotted on the y-axis versus magnitude of change (fold change) on the x-axis; significance threshold is set at p<0.05.

The online version of this article includes the following figure supplement(s) for figure 4:

**Figure supplement 1.** STRING analysis of proteins in RBCs of COVID-19 patients.

*Figure 4 continued on next page*

*Figure 4 continued*

**Figure supplement 2.** STRING analysis of positively correlated proteins with spheroechinocyte percentage, highlighting the main processes involving pathological RBCs in COVID-19.

### RBC morphological correlations with RBC content

Most correlations to both sphero-echinocytes and the fraction of pathological RBC shapes are observed with respect to RBC content (*Figure 4D*, *Figure 3—source data 1*). Negative correlations result with lactate, in contrast with the positive correlation in plasma, and mannitol, for which decreased levels are associated with reduced deformability of RBCs (*Burke et al., 1981*). Among the proteins is ALB, known to re-establish RBC discocyte shape in stored blood (*Reinhart et al., 2015*) and TF, which is reduced in COVID-19 plasma and is correlated with increased levels of CRP and IL-6 (*Claise et al., 2022*, *Figure 3—source data 1*). Tryptophan and the related metabolites formyl-kynurenine, kynurenine and serotonin negatively correlate with pathological RBC shapes, coherently with lower levels seen in patient RBCs (formyl-kynurenine, *Figure 4C*). Serotonin and dopamine result negatively correlated with the fraction of pathological RBC shapes. In contrast, positive correlations exist with several amino acids, such as methionine, tyrosine, lysine, histidine, threonine, and arginine catabolism products through the urea cycle, specifically citrulline, ornithine, arginine-succinate and urate. Creatine and creatinine levels also result positively correlated, further suggesting an enhanced arginine catabolism. These results highlight that RBC buffering activity to contrast the altered amino acid metabolism implies their morphological impairment. Finally, while formyl-kynurenine and kynurenine negatively correlate, positive correlations exist with anthranilate and picolinic acid, suggesting the ability of pathological RBCs to buffer kynurenine pathway metabolites. Positively correlated proteins are linked to different biological processes. Gas and ion transport-related proteins, such as carbonic anhydrase (CA2) and band3 (SLC4A1) are linked to pathological RBC shapes; a special highlight is a correlation with the expression of hemoglobin F subunits (HBG1 and HBG2). These data suggest a need to adjust oxygen transport, transporter activity, and pH buffering (chloride/bicarbonate ratios impact intracellular pH of RBCs) during the infection state (*Nemkov et al., 2018*). Other correlations are found with several proteasome-linked proteins, such as PSMB2, PSMB6, PSMA1, PSMC4, PSMC2 (*Figure 4—figure supplement 2*) with an ATP-dependent function, indicating high levels of protein degradation and regulation of amino acid metabolism in pathological RBCs. Indications of an enhanced antioxidant activity to cope with increased oxidative stress in such cells is highlighted by positive correlations with glutathione metabolism-related enzymes, such as glutathione S-transferase Mu 3 (GSTM3), glutathione reductase (GSR), which maintains reduced glutathione, and glucose-6-phosphate dehydrogenase (G6PD), the rate-limiting enzyme for the pentose-phosphate pathway (PPP), which is the only source for NADPH generation in RBCs, thus the main pathway for protection against oxidative stress. In addition, a positive correlation exists with peroxiredoxin 1 (PRDX1) that catalyzes the reduction of hydrogen peroxide. Further correlations are found with nicotinamide-phosphoribosyl transferase (NAMPT), which catalyzes the biosynthesis of NAD, and NADH-cytochrome b5 reductase 3 (CYB5R3), which reduces methemoglobin, the oxidated form of hemoglobin to the ferric state.

Together, these data show that morphological impairments observed in RBCs flowing in COVID-19 patient plasma are strongly related to markers of inflammation, oxidation, and hypoxia, which are associated with increased mortality risk (*D'Alessandro et al., 2021b*).

## Discussion

Our microfluidic analysis demonstrates a significantly increased number of pathological RBC shapes, such as sphero-echinocytes in the capillary flow of COVID-19 patients (PinP) (*Figure 2A–C*). These changes lead to an alteration of the microscale flow behavior since these RBCs are not able to adapt their shape to the imposed flow conditions compared to healthy controls (*Figure 2D*). Static images of the same suspensions confirm such morphological changes, where impaired RBCs can be visualized as sphero-echinocytes. Although patients show increased levels of fibrinogen (*Figure 1—source data 1*), which would favor an enhanced formation of RBC clusters (*Dasanna et al., 2022*), their RBCs do not aggregate into rouleaux but form disorganized structures. Hence, we hypothesize that the severe alterations of RBC shapes and deformability hinder the formation of RBC aggregates in stasis. However, COVID-19 RBCs are able to revert their shape to biconcave disks in stasis and healthy

croissant and slipper shapes in flow, resulting in a single-cell flow behavior comparable to control samples. Interestingly, the reversibility of sphero-echinocytes contradicts previous literature. Up to now, it was assumed that sphero-echinocytes cannot reverse to discocytes due to membrane loss through vesiculation that imbalances the structure of the lipid bilayer but this was not demonstrated mathematically (Lim H. W. et al., 2009). Marcel Bessis, one of the fathers of RBC classification based on their morphology, declared in his book 'Red Cell Shape' (*Bessis et al., 1973*) that sphero-echinocytes are not reversible to discocytes, without providing an experimental proof. This belief became so ingrained that it has never been argued. However, in previous work (*Ponder, 1948*), spherocytes, the very last transition stage of RBC before hemolysis, were described to be able to revert to biconcave disks as long as they have not reached the stage of prolytic spheres (*Ponder, 1948*). Moreover, it has been recently shown that morphological alterations concomitant to membrane shedding and acquisition of a spheroechinocytic phenotype, though apparently reversible in the present study, would still give rise to smaller RBCs of decreased volumes, which are preferentially cleared from the bloodstream via splenic sequestration (*Roussel et al., 2021*). Of note, echinocytes presence is not a specific feature of COVID-19 infection but has been previously observed in stasis in blood smears of septic patients (*Bateman et al., 2017*).

Patient plasma results in increased levels of inflammation and hypercoagulation markers, in accordance with previously published data (*Thomas et al., 2020b*). Particularly interesting due to its association with higher mortality risk is kynurenine, which decreases upon suspension of healthy control RBCs in COVID-19 plasma. Kynurenine levels are associated with hypoxia markers and interferon signaling (IFNG) (*D'Alessandro et al., 2021b*; *Galbraith et al., 2022*). The restoration of proteins and metabolites levels is acute and immediate and occurs at the expense of RBC shapes. Note that for the microfluidic experiments, RBCs must be diluted in control or patient plasma to obtain single-cell flow. This dilution could possibly impact the observed morphological changes of RBCs in patient plasma. With the used dilution, we see normal flow shapes for healthy RBC in autologous plasma, which are in agreement with previous studies in similar microchannels (*Guckenberger et al., 2018*; *Kihm et al., 2018*; *Recktenwald et al., 2022a*; *Recktenwald et al., 2022b*) and are therefore used as a basis to validate the chosen experimental setting. For the metabolomic and proteomic analyses, we use hematocrits close to the physiological ones. For these reasons, we believe that dilution is not a major reason for the described changes in RBCs and plasma in both types of samples (microfluidic and omics). The obtained results highlight a non-previously identified 'sponge-like' behavior of RBCs that is used to control plasma homeostasis. These influences and adaptations to the surrounding environment may be a specific feature of RBCs since they cannot synthesize new proteins and thus respond to needs through metabolic adjustments that also involve shape transformations. Proteomics and metabolomics data on RBCs indeed confirm that the differences between healthy and patient cells reflect the differences observed in their plasma content. A deeper investigation on morphologically impaired COVID-19 RBCs by correlation analysis results in their association with markers of inflammation, oxidation, and hypoxia in the plasma, features that act in a tight relation to overcome the infection.

Guanidinoacetate and ornithine correlations in patient RBCs suggest more activity of arginase that results in creatine production, which is used for ATP synthesis. It was previously seen that incubation of RBCs with arginine leads to the production of citrulline, ornithine, and urea, indicating an RBC-related enzymatic machinery for arginine metabolism (*Ramírez-Zamora et al., 2013*). The functional role of increased arginine catabolism may be to increase nitric oxide (NO) availability to respond to hypoxia, inducing vasodilation for an increased blood flow. Arginine supplementation increases T-cell proliferation and macrophage activity, thus it could contribute to enhance immune response (*D'Alessandro et al., 2021b*). As well, creatine leads to ATP synthesis, which increases during the immune response to boost leukocyte proliferation and activity. Moreover, tryptophan metabolism is involved in nicotinamide and NAD production. NAD is implicated in the glycolytic pathway and oxidative phosphorylation and its formation from tryptophan metabolism was demonstrated to boost macrophage phagocytic activity during immune response (*Minhas et al., 2019*). Dopamine has a role in the initiation of immune responses in lymphocytes (*Buttarelli et al., 2011*). The biosynthesis of serotonin and dopamine, which are negatively correlated with morphological RBC deviations, is thought to decrease in COVID-19 infection due to a co-expression and a functional link with Angiotensin I Converting Enzyme 2 (ACE2) (*Nataf, 2020*), the main receptor for SARS-CoV-2. Together, these data highlight the relevance of RBC buffering effect for immune cell activity regulation.

Positive correlations with plasma lactate levels are also linked to increased energy demand. Hypoxia causes a boost in glycolysis to produce and release more ATP that stimulates NO production and vasodilation, as well as 2,3-DPG that increases deoxyhemoglobin release of oxygen (*Nemkov et al., 2018*). ATP release also results in increased adenosine and xanthine in hypoxic plasma, as observed in our data. Adenosine levels remain high because PKA triggers a hypoxia-induced proteasome that degrades adenosine transporters (*Nemkov et al., 2018*), as we show from the correlation with several proteasome ATP-dependent proteins. More glycolysis results in less PPP for glutathione reduction, so less NADPH formation and presumably less antioxidant capacity of RBCs. Although the oxygen dissociation curve was not found different in COVID-19 patients (*Böning et al., 2021*) the presence of markers of hypoxia is expectable, considering that COVID-19 infection often results in lowered blood oxygen saturation, and in ECMO patients it is reduced until 82% (*Schmidt et al., 2013*). A reason for no differences in the oxygen dissociation curve may be due to methemoglobin, which increases hemoglobin oxygen affinity and may compensate the opposite effect of 2,3-DPG (*Böning et al., 2021*). Indeed, we found positive correlations with methemoglobin and hemoglobin F globin chains, whose expression is stimulated by hypoxia (*Simionato et al., 2022*). This highlights an effect of the infectious state also at the level of erythropoiesis.

Sphero-echinocytes show increased antioxidant activity since they are positively correlated with GSR, which catalyzes glutathione reduction, G6PD that is necessary for NADPH formation, and PRDX1, which reduces hydroperoxides. This response is associated with the infectious state since in septic patients proinflammatory cytokines such as TNF alpha promote ROS generation to destroy bacteria (*Bateman et al., 2017*). We could not find differences in, for example, levels of band 3 (AE1), to indicate oxidative stress-associated loss of cytoskeletal proteins, but it was reported oxidation of band 3, spectrin (SPTA1) and ankyrin (ANK1) (*Thomas et al., 2020a*), which, along with alterations of the lipid compartments, may alter RBC deformability. We see that the impaired deformability in COVID-19 cells is mostly reversible, thus not indicating damaged antioxidant mechanisms in RBCs, but rather increased oxidative stress caused by higher ROS generation during the infection that may be resolved by increasing RBC antioxidant defense.

The results of our study create the basis for possible clinical impacts: convalescent plasma transfusion was shown not to be associated with improved clinical outcomes (*Janiaud et al., 2021*). While plasma transfusions from healthy individuals may benefit RBC flow properties in COVID-19 patients that may potentially decrease RBC-related thrombotic risk (*Byrnes and Wolberg, 2017*), patient antibodies would be diluted, possibly compromising the efficacy of the immune response to the virus. Although the observed severe changes in RBC shapes could potentially act as a starting point for clinical studies, it is unclear whether a plasma or erythrocyte concentrate (EC) transfusion will be beneficial. Therefore, further studies are necessary to evaluate the mechanisms through which RBCs re-establish plasma equilibria, potentially decreasing patient mortality risk.

## Materials and methods

### Blood collection

Nine mL of blood is drawn in heparin tubes from five healthy volunteers under informed consent and 14 COVID-19 patients admitted at the Intensive Care Unit (ICU) at the Frankfurt University Hospital, seven of which with supported ventilation and six receiving extracorporeal membrane oxygenation (ECMO) (*Figure 1—source data 1*). The study is performed according to the Declaration of Helsinki and under the approval of the local ethics committee (reference #20–643, #20–982). Healthy and patient blood tubes are transported at room temperature and processed after 2 hr from blood drawing.

### Sample preparation

Blood is leukodepleted (Pall's Acrodisc PSF syringe filters, Pall Corporation, New York, NY) and centrifuged at 1500×g for 5 min to separate RBCs and plasma. RBCs are then suspended in PBS (Gibco PBS, Thermo Fisher, Bremen, Germany) and the centrifugation and washing steps are repeated three times. Plasma fraction is centrifuged at 5000×g for 5 min to assure the removal of platelets, then used to resuspend washed RBCs in autologous plasma at a final hematocrit of 0.5%. Additionally, patient RBCs are suspended in blood group-matching control plasma and control RBCs in patient plasma,

obtaining four sample groups; (i) control RBCs in control plasma (CinC), (ii) patient RBCs in patient plasma (PinP), (iii) control RBCs in patient plasma (CinP), and (iv) patient RBCs in control plasma (PinC) (*Figure 1A*).

## Microfluidic setup

A microfluidic channel with a rectangular cross-section of 8 μm height (H) and 11 μm width (W) is used to pump RBC suspensions as described (*Recktenwald et al., 2022a*). RBC flow x-y-plane is recorded with a frame rate of up to 400 fps using a pressure drop range between 100 mbar and 1000 mbar. The resulting cell velocities $U$ are in the range of 0.7 mm s$^{-1}$ and 7 mm s$^{-1}$, similar to the flow in the microvascular network (*Pries and Secomb, 2008*; *Secomb, 2017*). Based on these velocities, we estimate the wall shear rate as $\dot{\gamma}_w = 2U/(H/2)$, which is in the range of approximately 350 s$^{-1}$ to 3500 s$^{-1}$. This corresponds to a wall shear stress $\tau_w$ of 0.42 Pa to 4.2 Pa, based on a plasma viscosity of approximately $\eta = 1.2\,\mathrm{mPa\,s}$.

## Flow analysis

Velocity, lateral cell position in the y-direction, and projection area of each cell are determined using a customized python script. Per patient/donor an average of 3317 individual cells (between 1199 and 5496 cells) are examined. Based on the RBC velocity, the distribution of RBC lateral y-position is used as a characteristic indicator of the single-cell flow in such confined microchannels (*Kihm et al., 2018*; *Guckenberger et al., 2018*; *Recktenwald et al., 2022a*; *Recktenwald et al., 2022b*). At a given mean velocity, the distribution of the absolute value of the cell lateral position in y-direction, normalized by the channel width, is determined and the corresponding probability density function (pdf) is calculated. At low velocities, RBCs preferentially form axisymmetric croissants that flow in the channel center, while slipper-shaped RBCs that emerge at higher velocities flow at an off-centered equilibrium position (*Figure 1—figure supplement 1*). We quantify the difference between pathological and healthy RBC flow by the so-called y-deviation, which relates the pdfs of the y-distributions of a given sample to the average distribution of healthy controls at specific cell velocities (*Figure 1—figure supplement 1*).

Additionally, we use a convolutional neural network (CNN) for a non-biased RBC shape analysis. The CNN consist of an image input layer, several subsequent convolution stages, and an output layer, as previously described (*Kihm et al., 2018*; *Recktenwald et al., 2022b*). The network is trained with 1000 representative images of each healthy and pathological shape (*Figure 1C and D*) that were manually classified before.

## Omics sample collection

Four hundred μL packed RBCs is suspended in 600 μL autologous or allogeneic compatible plasma for 20 min at 4°C resulting in a total sample volume of 1 mL with a hematocrit of 40%. Plasma is collected according to a modified Folch extraction method (*Burnum-Johnson et al., 2017*) and the three fractions obtained (lipid, metabolite and protein-enriched) are stored in liquid nitrogen until analysis, as well as packed RBCs after suspension in autologous or allogeneic plasma. Samples are collected from three healthy controls and eight patients, of which four were with supported ventilation and four on ECMO.

## Ultra-high-pressure liquid chromatography-mass spectrometry (UHPLC-MS) metabolomics

A volume of 50 μL of frozen RBC aliquots is extracted in 950 μL methanol:acetonitrile:water (5:3:2, v/v/v). Samples are vortexed and insoluble material pelleted as described (*D'Alessandro et al., 2021a*). Analyses are performed using a Vanquish UHPLC coupled online to a Q Exactive mass spectrometer (Thermo Fisher, Bremen, Germany). Samples are analyzed using a 3-min isocratic condition or a 5, 9, and 17 min gradient as described (*Nemkov et al., 2019*). Solvents are supplemented with 0.1% formic acid for positive mode runs and 1 mM ammonium acetate for negative mode runs. MS acquisition, data analysis, and elaboration are performed as described (*Nemkov et al., 2019*). Additional analyses, including untargeted analyses and Fish score calculation via MS/MS, are calculated against the ChemSpider database with Compound Discoverer 2.0 (Thermo Fisher, Bremen, Germany).

## Proteomics

Proteomics analyses are performed via filter aided sample preparation (FASP) digestion and nano UHPLC-MS/MS identification (TIMS TOF Pro 2 Single Cell Proteomics, Bruker Daltonics, Bremen, Germany), as previously described (*Issaian et al., 2021*).

## Statistical analyses

Graphical representations and statistical analyses by T-test, repeated measures ANOVA or Kruskal-Wallis test are performed with GraphPad Prism (GraphPad Software, Inc, La Jolla, CA), MATLAB (MathWorks, Natick, MA), and MetaboAnalyst 5.0. Spearman's rank correlation coefficient analysis is applied between the microfluidic and omics data. Both correlation data and omics raw data are included in *Figure 3—source data 1*.

## Acknowledgements

This work was supported by the European Union's Horizon 2020 research and innovation program under the Marie Skłodowska-Curie grant agreement No 860436 – EVIDENCE and by the Deutsche Forschungsgemeinschaft (DFG, German Research Foundation) in the framework of the research unit FOR 2688 'Instabilities, Bifurcations and Migration in Pulsatile Flows' WA 1336/13–1 and in the 'Open Access Publication Funding' program with support by the Saarland University. We would like to thank Annett Wilken-Schmitz for the organizational support and collection of blood samples.

## Additional information

### Competing interests

Steffen M Recktenwald, Greta Simionato, Lars Kaestner, Stephan Quint: Though unrelated to the contents of this manuscript, the authors declare that SMR, GS, LK, and SQ are co-founders of Cysmic GmbH. Marcelle GM Lopes: Affiliated with Cysmic GmbH; the author has no financial interests to declare. Angelo D'Alessandro: Though unrelated to the contents of this manuscript, AD is a founder of Omix Technologies Inc and Altis Biosciences LLC, and a scientific advisory board member for Hemanext Inc and Forma Therapeutics. The other authors declare that no competing interests exist.

### Funding

| Funder | Grant reference number | Author |
| --- | --- | --- |
| Horizon 2020 Framework Programme | 860436 | Marcelle GM Lopes<br>Christian Wagner<br>Lars Kaestner<br>Stephan Quint |
| Deutsche Forschungsgemeinschaft | WA 1336/13-1 | Steffen M Recktenwald<br>Greta Simionato<br>Christian Wagner |
| Universität des Saarlandes | Open Access Publication Funding | Steffen M Recktenwald<br>Greta Simionato |

The funders had no role in study design, data collection, and interpretation, or the decision to submit the work for publication.

### Author contributions

Steffen M Recktenwald, Greta Simionato, Conceptualization, Data curation, Formal analysis, Investigation, Methodology, Writing – original draft, Writing – review and editing; Marcelle GM Lopes, Data curation, Writing – review and editing; Fabia Gamboni, Monika Dzieciatkowska, Data curation, Formal analysis, Writing – review and editing; Patrick Meybohm, Resources, Writing – review and editing, Provided blood samples and clinical data interpretation; Kai Zacharowski, Resources, Writing – review and editing, Provided blood samples and clinical data interpretation; Andreas von Knethen, Resources, Writing – review and editing, Provided blood samples and clinical data interpretation; Christian Wagner, Resources, Funding acquisition, Writing – review and editing; Lars Kaestner,

Conceptualization, Writing – review and editing; Angelo D'Alessandro, Resources, Formal analysis, Supervision, Funding acquisition, Writing – review and editing; Stephan Quint, Conceptualization, Data curation, Supervision, Investigation, Methodology, Writing – review and editing

### Author ORCIDs
Steffen M Recktenwald  http://orcid.org/0000-0003-1235-1521
Greta Simionato  http://orcid.org/0000-0001-5452-2275
Marcelle GM Lopes  http://orcid.org/0000-0002-9668-5504
Fabia Gamboni  http://orcid.org/0000-0002-0855-9138
Patrick Meybohm  http://orcid.org/0000-0002-2666-8696
Kai Zacharowski  http://orcid.org/0000-0002-0212-9110
Andreas von Knethen  http://orcid.org/0000-0002-5831-0365
Christian Wagner  http://orcid.org/0000-0001-7788-4594
Lars Kaestner  http://orcid.org/0000-0001-6796-9535
Stephan Quint  http://orcid.org/0000-0003-4412-7559

### Ethics
Human subjects: The study is performed according to the Declaration of Helsinki and under the approval of the local ethics committee (reference \#20-643, \#20-982) from five healthy volunteers and 14 COVID-19 patients under informed consent.

### Decision letter and Author response
Decision letter https://doi.org/10.7554/eLife.81316.sa1
Author response https://doi.org/10.7554/eLife.81316.sa2

## Additional files

### Supplementary files
• MDAR checklist

### Data availability
All data generated or analysed during this study are included in the manuscript and supporting files. Source data files are provided for Figures 3 and 4.

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
