## [Editor Report]

This report illustrates a comprehensive account detailing the marked alteration of red blood cell (RBC) morphology that occurs with COVID-19 infection. A particularly important result is the observation that RBC morphology is dramatically affected by plasma from COVID-19 patients and reversible with plasma from healthy donors. The claims of the manuscript are well supported by the data, and the approaches used are thoughtful and rigorous. The results are important for consideration of the broader pathophysiology of COVID-19, particularly with regard to the impact on vascular biology and will be of interest to the readership of *eLife*.

---

## [Decision Letter]

**Decision letter after peer review:**

Thank you for submitting your article "Cross-talk between red blood cells and plasma influences blood flow and omics phenotypes in severe COVID-19" for consideration by *eLife*. Your article has been reviewed by 2 peer reviewers, and the evaluation has been overseen by a Reviewing Editor and Mone Zaidi as the Senior Editor. The following individuals involved in the review of your submission have agreed to reveal their identity: Michael Graham (Reviewer #1); Chaouqi Misbah (Reviewer #2).

*Reviewer #1 (Recommendations for the authors):*

My question in the public review about the role of dilution in the observations is my primary reservation about the publication of this interesting manuscript in its current state.

*Reviewer #2 (Recommendations for the authors):*

Overall, the study is relevant to the community working on the multifactorial effect of Covid-19. The paper is well written, and it deserves publication after the authors have addressed the following points

1) I was a bit surprised by the fact that both PinP and CinP RBCs do not show clusters. It seems to mean that either (i) plasma of patients is altered significantly so that fibrinogen (among other molecules) has been altered, in a way that depletion force is weak (if we admit depletion mode as the mechanism responsible for clustering) (ii) or RBCs membrane is altered, if one admits the bridging mode to be responsible for aggregation. These questions need further discussion, and experimental support, if possible.

2) Often clustering is associated with fibrinogen, but the role of albumin, as well as Igg antibodies, is often not stressed enough. It is reported here that in control plasma albumin is more abundant. Would that explain partially the point raised in 1) above? Did the authors examine Igg in control and compared it to patients' plasma? At least it would be nice to discuss these issues in the revised version.

3) As explained above in 1) clusters are absent in patient plasma. At the same time, it is known that blood occlusion is a major cause of patients' death. Of course, many other factors can enter into play for blood occlusion. Does this study inform us of the role of this finding in the formation of blood occlusion? Maybe the interaction among RBCs is so weak that a small stress may dissociate them. Can the authors comment on documented origins of vessel occlusions?

4) I have a naive question: is it known, or not, if cytokines have any effect on RBCs shapes, on plasma, or on collective effects, such as firm cluster formation?

---

## [Author Response]

Reviewer #2 (Recommendations for the authors):Overall, the study is relevant to the community working on the multifactorial effect of Covid-19. The paper is well written, and it deserves publication after the authors have addressed the following points1) I was a bit surprised by the fact that both PinP and CinP RBCs do not show clusters. It seems to mean that either (i) plasma of patients is altered significantly so that fibrinogen (among other molecules) has been altered, in a way that depletion force is weak (if we admit depletion mode as the mechanism responsible for clustering) (ii) or RBCs membrane is altered, if one admits the bridging mode to be responsible for aggregation. These questions need further discussion, and experimental support, if possible.

We thank the reviewer for raising this important question. We measured the fibrinogen levels of the patients (see Figure 1-source data 1: a supplementary table containing the patient information), which exhibit on average higher values than for healthy controls (patients: mean: 427 mg/dL; min: 276 mg/dL; max: 624 mg/dL). Increased fibrinogen levels lead to increased clustering and sedimentation of RBCs (e.g., Darras et al., PRL 2022, https://doi.org/10.1103/PhysRevLett.128.088101; and Dasanna et al., PRE 2022, https://doi.org/10.1103/PhysRevE.105.024610). However, here we find that patients show a reduced number of RBC clusters in stasis (Figure 1B and Figure 1—figure supplement 2) as compared to healthy controls, although they exhibit on average increased fibrinogen levels. Therefore, we think that the strong alterations in the RBC shapes of patients (e.g., the presence of sphero-echinocytes) is the driving force that hinders the formation of RBC aggregates, irrespective of the aggregation mode (depletion/bridging).

We added a corresponding sentence in the Discussion section regarding the RBC shape and clustering in stasis.

2) Often clustering is associated with fibrinogen, but the role of albumin, as well as Igg antibodies, is often not stressed enough. It is reported here that in control plasma albumin is more abundant. Would that explain partially the point raised in 1) above? Did the authors examine Igg in control and compared it to patients' plasma? At least it would be nice to discuss these issues in the revised version.

We measured the albumin levels of the patients (see Figure 1-source data 1: a supplementary table containing the patient information), which exhibit smaller values than for healthy controls (patients: mean:2.9 mg/dL; min: 1.8 mg/dL; max: 3.3 mg/dL). We did not measure immunoglobulins levels. Although albumin (as well as immunoglobulins like IgG and IgM) can play a role in clustering, fibrinogen is a more efficient aggregation-promoting agent of RBCs. However, as stated in the response to comment 1, reviewer 2, we hypothesize that the main reason why RBCs in patient plasma do not form clusters is the presence of sheroechinocytes that prevent the cells from forming RBC aggregates.

3) As explained above in 1) clusters are absent in patient plasma. At the same time, it is known that blood occlusion is a major cause of patients' death. Of course, many other factors can enter into play for blood occlusion. Does this study inform us of the role of this finding in the formation of blood occlusion? Maybe the interaction among RBCs is so weak that a small stress may dissociate them. Can the authors comment on documented origins of vessel occlusions?

We thank the reviewer for this interesting comment. Under low shear rates and deformations, RBCs form rouleaux that can form larger clusters. However, under flow conditions, as is in the circulatory system, rouleaux structures would break up, and hence, such aggregates are not relevant for blood occlusions, as stated by the reviewer. Nevertheless, patient RBCs exhibit a strong impairment in cell deformability. Therefore, RBCs are not able to adapt their shape to the flow conditions, which leads to large y-deviations in our study (Figure 2D and Figure 1—figure supplement 1). Since a decreased deformability can result in complications such as damage to the endothelial cells lining the blood vessel walls (e.g., Caruso et al., 2022), we think that the severe shape and deformability impairment of patient RBCs contributes to the severity of COVID-19.

4) I have a naive question: is it known, or not, if cytokines have any effect on RBCs shapes, on plasma, or on collective effects, such as firm cluster formation?

We thank the reviewer for this comment and it is indeed not naïve. As cytokines affect various blood cells, we suspect there is a complex effect of cytokines on the RBC shape. However, these concrete effects are beyond the scope of the current manuscript.